# Two-photon-absorbing ruthenium complexes enable near infrared light-driven photocatalysis

Guanqun Han[1], Guodong Li[1], Jie Huang[2], Chuang Han[1], Claudia Turro[2✉] & Yujie Sun [1✉]

One-photon-absorbing photosensitizers are commonly used in homogeneous photocatalysis which require the absorption of ultraviolet (UV) /visible light to populate the desired excited states with adequate energy and lifetime. Nevertheless, the limited penetration depth and competing absorption by organic substrates of UV/visible light calls upon exploring the utilization of longer-wavelength irradiation, such as near-infrared light ($\lambda_{irr} > 700$ nm). Despite being found applications in photodynamic therapy and bioimaging, two-photon absorption (TPA), the simultaneous absorption of two photons by one molecule, has been rarely explored in homogeneous photocatalysis. Herein, we report a group of ruthenium polypyridyl complexes possessing TPA capability that can drive a variety of organic transformations upon irradiation with 740 nm light. We demonstrate that these TPA ruthenium complexes can operate in an analogous manner as one-photon-absorbing photosensitizers for both energy-transfer and photoredox reactions, as well as function in concert with a transition metal co-catalyst for metallaphotoredox C–C coupling reactions.

[1] Department of Chemistry, University of Cincinnati, Cincinnati, OH, USA. [2] Department of Chemistry & Biochemistry, The Ohio State University, Columbus, OH, USA. ✉email: turro.1@osu.edu; yujie.sun@uc.edu

The rapid advancement in homogeneous photocatalysis has primarily relied on the employment of one-photon-absorbing photosensitizers, such as ruthenium poly-pyridyl complexes[1,2], cyclometalated iridium complexes[3], and organic dyes[4–6]. In all of these examples, the photosensitizers are excited by either ultraviolet (UV) or visible light (in most cases blue light) to their excited states with competent energy and lifetime to drive organic transformations[7]. It is very rare to employ near-infrared (NIR) light in conventional one-photon-absorbing photocatalysis, because of the low energy and short lifetime of the excited states that are obtained under NIR light excitation. However, the utilization of UV/visible photons presents several intrinsic limitations such as poor penetration through reaction media[8,9], competing absorption by species involved in reaction[10–12], and incompatibility against substrates bearing light-sensitive functionalities[13], as well as limited coverage of the solar spectrum. Therefore, the exploration of NIR light-driven photocatalysis is critical to avoid most of the aforementioned drawbacks of UV/visible light irradiation.

Because of the lower energy of NIR photons relative to UV/visible photons, energy must be harvested from two (or more) NIR photons to produce the desired excited state. One strategy to achieve this scenario is triplet-triplet annihilation upconversion. Indeed, it was recently demonstrated that NIR light-absorbing sensitizers can be used to activate suitable annihilators for homogeneous photocatalysis[14–16]. Nevertheless, the success of triplet-triplet annihilation upconversion relies on the fine state energy matching and spatial interaction of sensitizers and annihilators, potentially limiting its use due to its complex nature and the availability of suitable sensitizer/annihilator pairs. An alternative and more straightforward strategy for NIR photocatalysis is through direct two-photon absorption (TPA), wherein a chromophore is capable of simultaneously absorbing two photons in a single step to populate the desired excited state. Even though the TPA phenomenon was predicted by Göppert-Mayer in 1931[17] and utilized in fields like bioimaging[18,19] and photodynamic therapy (PDT)[20,21], its application in homogeneous photocatalysis remains very much underexplored, likely due to the small TPA cross sections ($\sigma_2$) of most photosensitizers suitable for photocatalysis[22]. For instance, the widely used photosensitizer $[Ru(bpy)_3]^{2+}$ (bpy = 2,2'-bipyridyl) was reported to possess a TPA cross section of only 4.3 GM at 880 nm (1 GM = $10^{-50}$ cm⁴ s photon⁻¹)[23], rendering its potential to function as a TPA photosensitizer in the NIR region negligible. Only a handful of TPA systems have been investigated for catalysis, however visible light and/or intense lasers are required as the light source[24–28]. It should be noted that several multi-photon-absorbing catalytic systems have also been reported, however they either require consecutive absorption of photons by two different species or follow the triplet-triplet annihilation upconversion mechanism[29–38]. To the best of our knowledge, there has been no reports on simultaneous two-photon absorption by one molecule for organic reactions using inexpensive NIR LEDs ($\lambda_{irr} > 700$ nm).

Along the continuous efforts in designing novel chromophores with enhanced two-photon absorption, it was found that strong intramolecular charge transfer and superpolarizability may yield high TPA cross section[39]. For instance, stilbene was measured with $\sigma_2 = 12$ GM at 514 nm and electron-donating substituents substantially increased the $\sigma_2$ value[40]. These early findings raised the interest in developing organic TPA compounds with symmetrical donor–π–acceptor structures. In the meantime, transition metal complexes coordinated with elongated π-conjugation ligands have also been explored for TPA[41]. For instance, Ru-based TPA complexes have been gradually developed as PDT agents[42] and photo-activable compounds[43]. Unfortunately, the TPA cross sections of most Ru complexes are extremely small (<10 GM)[44–46], making their utilization in NIR photocatalysis impractical.

We were inspired by a few studies showing that octupolar bipyridyl metal complexes coordinated with bisstyryl-substituted bpy ligands exhibit superpolarizability and TPA cross sections two orders of magnitude higher than that of $[Ru(bpy)_3]^{2+}$ in the NIR region[47–49], some of which were also demonstrated as competent PDT agents upon excitation at 700–800 nm[50]. Hence, we became interested in exploring the possibility of utilizing Ru complexes coordinated with extended π-conjugation ligands for NIR photocatalysis with TPA excitation. Herein, we report a group of Ru complexes ligated by 4,4'-bisstyryl-2,2'-bipyridine (bpyvp) and its close analogues, that are able to drive various organic reactions in a manner akin to conventional one-photon-absorbing photosensitizers, but absorbing two photons per step in the NIR region (Fig. 1a). Both energy-transfer reactions (e.g., ¹O₂ reactions) and photoredox reactions (e.g., hydrodehalogenation, C–H cyanation, Ni-catalyzed allylation of aldehydes) can be accomplished by these Ru complexes with excellent yields upon irradiation at 740 nm under ambient conditions. By tuning the ligand ratio between bpy and bpyvp-type ligands, as well as installing electronic-donating (e.g., -OMe) or -withdrawing (e.g., –F) substituents at the para positions of the terminal phenyl groups in bpyvp (Fig. 1b), together with theoretical computation,

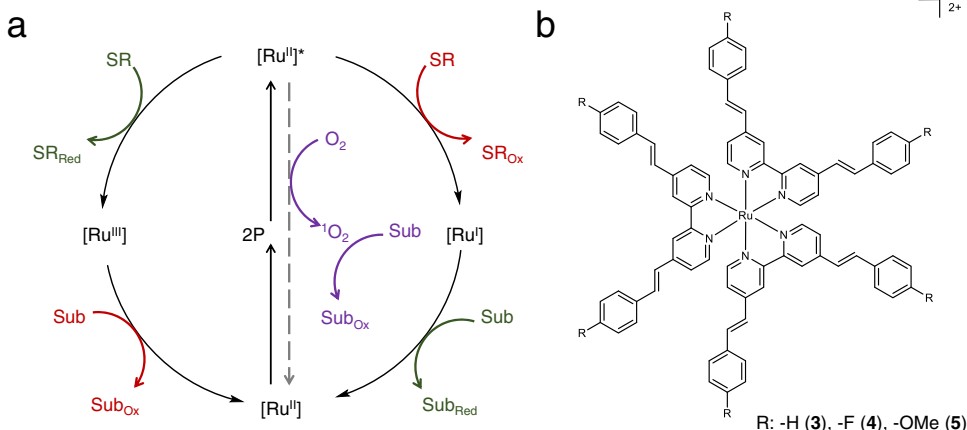

**Fig. 1 Photocatalytic schemes and molecular structures of representative Ru TPA photosensitizers. a** Possible reaction pathways for two-photon-absorbing ruthenium photosensitizers following both energy- and electron-transfer processes. [Ru] ruthenium photosensitizer, SR sacrificial reagent, Sub substrate, Ox oxidation, Red reduction. **b** Molecular structures of Ru TPA complexes **3**, **4**, and **5**.

**Table 1 Photophysical and electrochemical properties of complexes 1–5 compared with [Ru(bpy)$_3$]$^{2+}$.**

| Complex | $\lambda_{abs}$/nm ($\varepsilon$/M$^{-1}$ cm$^{-1}$ 10$^{-3}$) | $\lambda_{em}$/nm | $\tau$/ns | E$_{1/2}$ /V vs. Fc$^{+/0a}$ |
|---|---|---|---|---|
| [Ru(bpy)$_3$]$^{2+}$ | 288 (69.3), 450 (14.6) | 619 | 950[b] | +0.88, −1.73, −1.91, −2.18 |
| **1** | 289 (90.0), 325 (52.3), 465 (24.9) | 656 | – | +0.77, −1.64, −1.88, −2.12 |
| **2** | 290 (96.9), 330 (91.6), 475 (35.7) | 664 | – | +0.73, −1.63, −1.79, −2.08 |
| **3** | 290 (107.7), 328 (142.1), 488 (48.8) | 667 | 431 (127) | +0.67, −1.62, −1.76, −1.95 |
| **4** | 290 (97.8), 330 (121.7), 488 (47.1) | 668 | 628 (131) | +0.68, −1.62, −1.76, −1.95 |
| **5** | 305 (97.2), 365 (143.2), 490 (60.8) | 673 | 877 (127) | +0.62, −1.67, −1.81, −2.01 |

$\lambda_{abs}$ Absorption maximum, $\varepsilon$ Extinction coefficient, [b]$\lambda_{em}$ Emission maximum, [c]$\tau$ Excited state lifetime in deaerated (or aerated) CH$_3$CN.
[a]Measured in deaerated DMF with 0.1 M LiClO$_4$ as the supporting electrolyte.
[b]From ref. [51].

we are able to rationalize the relationship between the electronic structures of these photosensitizers and their TPA photocatalytic performance, providing guidance for the development of improved TPA photosensitizers for NIR photocatalysis.

## Results and discussion

**Synthesis, physical characterization, and theoretical calculation.** Following the reported procedure[50], the parent ligand 4,4'-bisstyryl-2,2'-bipyridine (bpyvp-H) was successfully synthesized with an overall yield of 80%. By controlling the ratio between bpyvp-H and the corresponding Ru precursor, we were able to obtain heteroleptic [Ru(bpy)$_2$(bpyvp-H)]$^{2+}$ (**1**), [Ru(bpy)(bpyvp-H)$_2$]$^{2+}$ (**2**), and homoleptic [Ru(bpyvp-H)$_3$]$^{2+}$ (**3**) in decent yields. The fluorine- and methoxyl-substituted ligands, bpyvp-F and bpyvp-OMe, were synthesized in a similar fashion as bpyvp-H. Hence, two additional homoleptic Ru complexes, [Ru(bpyvp-F)$_3$]$^{2+}$ (**4**) and [Ru(bpyvp-OMe)$_3$]$^{2+}$ (**5**), were also synthesized (Supplementary Figs. 1–9).

The photophysical and electrochemical properties of **1–5** are summarized in Table 1 and [Ru(bpy)$_3$]$^{2+}$ is included for comparison. It is apparent that as the number of bpyvp-H ligands is increased in these Ru complexes, the metal-to-ligand charge transfer (MLCT) band red shifts from 465 nm in **1** (Supplementary Fig. 10), 475 nm in **2** (Supplementary Fig. 11), to 477 nm in **3** (Supplementary Fig. 12), together with a two-fold increase in the extinction coefficient ($\varepsilon$). The presence of the fluorine substituents in **4** does not substantially alter its absorption (Supplementary Fig. 13) relative to that of **3**, while a further red shift in the MLCT band and increased $\varepsilon$ are observed for **5** with methoxy substituents (Fig. 2b and Supplementary Fig. 14). In addition to the bpy-centered $\pi\pi^*$ transition located around 290 nm, **1–5** present an additional absorption feature in 300–400 nm, whose $\varepsilon$ values increase approximately linearly with the number of bpyvp-type ligands (52–143 × 10$^3$ M$^{-1}$ cm$^{-1}$). Therefore, these absorption bands are tentatively assigned to bpyvp-based $\pi\pi^*$ and/or intraligand charge transfer transitions. Supplementary Figs. 10–14 also present the emission of **1–5** with maxima at 660–670 nm, along with the corresponding excitation spectra. The good overlap between the excitation and absorption spectra of each complex confirms that the each emission arises from the target Ru complexes.

Cyclic and linear sweep voltammograms of **1–5** (Supplementary Figs. 15–19) reveal four reversible redox couples, one anodic feature within the +0.62 to +0.77 V vs Fc$^{+/0}$ (Fc: ferrocene) range and three reduction couples between −1.62 and −2.12 V vs Fc$^{+/0}$. A linear relationship between the peak current and the square root of the scan rate was obtained for each complex, confirming the molecular nature free diffusion of the redox-active species in solution (Supplementary Figs. 15c–19c). The negative shifts of the anodic features of **1–5** relative to that of [Ru(bpy)$_3$]$^{2+}$ (+0.80 V vs Fc$^{+/0}$) can be attributed to the enhanced electron donation from the bpyvp-type ligands. As expected, the extended $\pi$ conjugation of the bpyvp-type ligands also results in lower-lying LUMOs

(lowest unoccupied molecular orbitals), thus leading to anodic shifts of their first reduction potentials as compared to bpy. Indeed, density functional theory (DFT) calculation results (Supplementary Tables 1–5) support the distribution of LUMOs primarily located on the bpyvp-type ligands instead of bpy (see the LUMOs of **1** and **2** in Supplementary Tables 1 and 2, respectively) and the energy trend of each LUMO in **1–5** is consistent with their first reduction couples, wherein **5** coordinated with three bpyvp-OMe shows a higher-lying LUMO (Fig. 2a) than that of **3** and thus more negative first reduction potential for the former, −1.67 V vs Fc$^{+/0}$. The calculated HOMOs of **1–5** are primarily centered on the Ru 4d orbitals with slight contribution from the bpyvp-type ligands, as evidenced by the HOMO of **5** shown in Fig. 2a.

Time-dependent DFT (TD-DFT) calculations (Supplementary Tables 6–10) assisted us in understanding the singlet excited states (S$_n$) of each complex. Taking **5** as an example, its S$_1$ state was calculated at 551 nm and its electron density difference map (EDDM) further confirms its MLCT character with all three bpy components in bpyvp-OMe acting as the charge accepting units. In addition, the prominent S$_8$ state ($f = 0.50$) of **5** appearing at 512 nm aligns very well with the MLCT band in its UV–vis absorption spectrum (Fig. 2b). Transitions with mixed MLCT and LCCT (ligand-centered charge transfer) character were computed for the singlet excited states of **5** with energies between 350 and 450 nm. For instance, the EDDMs of two dominant states, S$_{13}$ ($f = 0.89$) and S$_{22}$ ($f = 0.81$), are displayed in Fig. 2b, clearly presenting the contribution of both MLCT and LCCT transitions. Similar singlet excited states of mixed nature were also obtained from the TD-DFT calculation results of complexes **1–4**, wherein those bpyvp-type ligands, instead of bpy, more likely acted as the electron density-receiving units, likely due to their extended $\pi$-conjugation structures.

Despite the general similarities in one-photon photophysical and electrochemical properties of complexes **1–5** and [Ru(bpy)$_3$]$^{2+}$[51], these complexes are strikingly different in their ability to undergo two-photon absorption (TPA). As shown in Fig. 2c, upon excitation at 480 nm, the transient absorption spectra of **5** show the characteristic MLCT bleach at ~500 nm and excited state absorption in the 400–450 nm range and beyond 550 nm. Importantly, when exciting with 800 nm, a wavelength that complex **5** does not absorb (Fig. 2b), a nearly identical transient absorption spectrum was obtained (Fig. 2d). Furthermore, a linear relationship is observed when the transient absorption signal at 500 nm is plotted versus the square of the excitation power ($\lambda_{irr} = 800$ nm), shown in the inset of Fig. 2d. Similar results were obtained for the other complexes (Supplementary Figs. 20–22). The fact that analogous transient absorption spectra were collected by excitation at both 480 and 800 nm, together with a linear relationship between the optical density in the transient absorption and the square of excitation power at 800 nm, unambiguously confirm that these Ru complexes indeed possess TPA capability and identical excited states are populated when excited at either 480 or 800 nm.

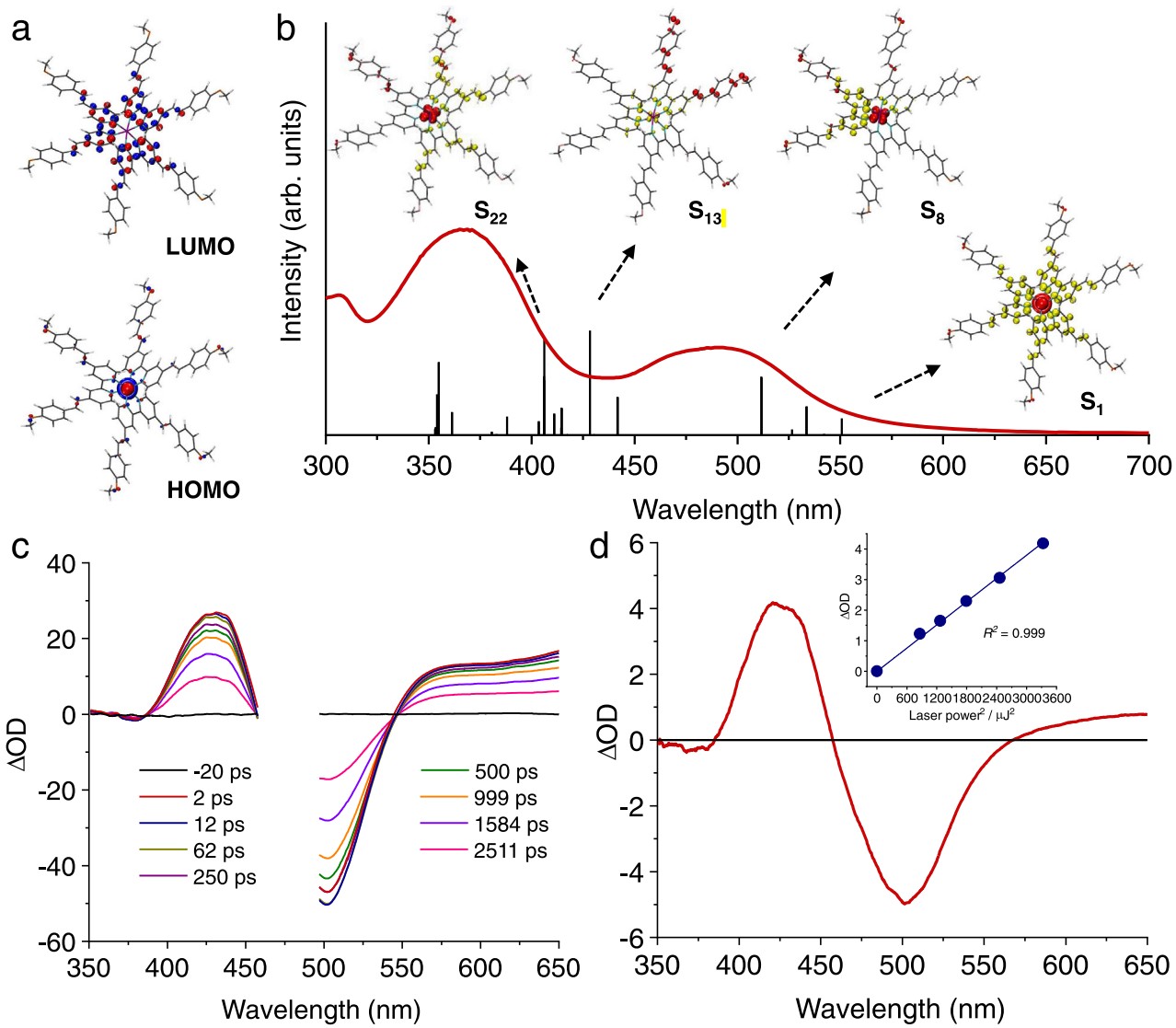

**Fig. 2 Calculated frontier orbitals, absorption, and transient absorption spectra of 5. a** Calculated HOMO and LUMO of **5** (Isovalue = 0.04). **b** Calculated singlet electronic transitions of **5** together with the selected EDDMs (Isovalue = 0.04) of $S_1$, $S_8$, $S_{13}$, and $S_{22}$ states (red indicates electron decrease and yellow indicates electron increase). The experimental absorption spectrum (red line) is also included for comparison. **c**, **d** Transient absorption spectra of **5** under excitation at 480 nm (**c**) and 800 nm (**d**). The inset of (**d**) plots the relationship between the transient absorption signal density at 500 nm and square of the excitation light power.

**Photocatalytic $^1O_2$-driven energy transfer reactions under NIR light irradiation.** The excited state lifetimes of the homoleptic complexes **3**, **4**, and **5** in deaerated $CH_3CN$ were measured at 431, 628, and 877 ns, respectively (Table 1 and Supplementary Figs. 23–25), which all decreased to ~130 ns in air, consistent with the excited state energy transfer to $O_2$ to generate reactive singlet oxygen $^1O_2$. Given this reactivity, we sought to utilize photocatalytic C–N coupling of benzyl amine as a model reaction to evaluate the photocatalytic performance of these complexes under NIR irradiation. Commercially inexpensive LEDs ($\lambda_{irr}$ = 740 nm, 8.18 W) were employed as the light source in all the following photocatalysis experiments. It is known that benzyl amine can be oxidized by $^1O_2$ to produce N-benzylidenebenzylamine[52,53]. As shown in Fig. 3a and Supplementary Figs. 26–30, a dramatic difference in reaction rate was detected for **1**–**5**. It is apparent that the increasing number of bpyvp is beneficial for the overall photocatalysis, in that the reaction rate follows the order **1** < **2** < **3**. Only a 16% yield of N-benzylidenebenzylamine after 2 h irradiation was realized for **1**, while benzyl amine was nearly fully

converted to N-benzylidenebenzylamine within 100 and 50 min using **2** and **3** as the photosensitizer, respectively. Using **3** as the standard photocatalyst, a series of control experiments (Supplementary Fig. 31) prove that light irradiation, $O_2$, and photosensitizer are all crucial for the success of this reaction (entries 2, 3, and 4 in Fig. 3b). When the reaction is conducted in air, a moderate yield of 43% was obtained within 30 min (entry 5 in Fig. 3b), lower than that measured under $O_2$ (89%). It is interesting to find that **5** is more effective in initiating the benzyl amine C-N coupling than **3**, with a yield of 96% for **5** within 30 min upon 740 nm irradiation as compared to **3**, which achieved an 89% yield. This difference can be attributed to the longer excited state lifetime of **5**, expected to generate a greater amount of $^1O_2$, as well as its higher TPA cross section. In contrast, complex **4**, possessing fluorine substituents, only reached a yield of 42% under the same conditions. These results imply that better electron donation from the terminal phenyl group to the metal center, and hence greater intra-molecular charge transfer, can be beneficial for NIR TPA photocatalytic performance. Given

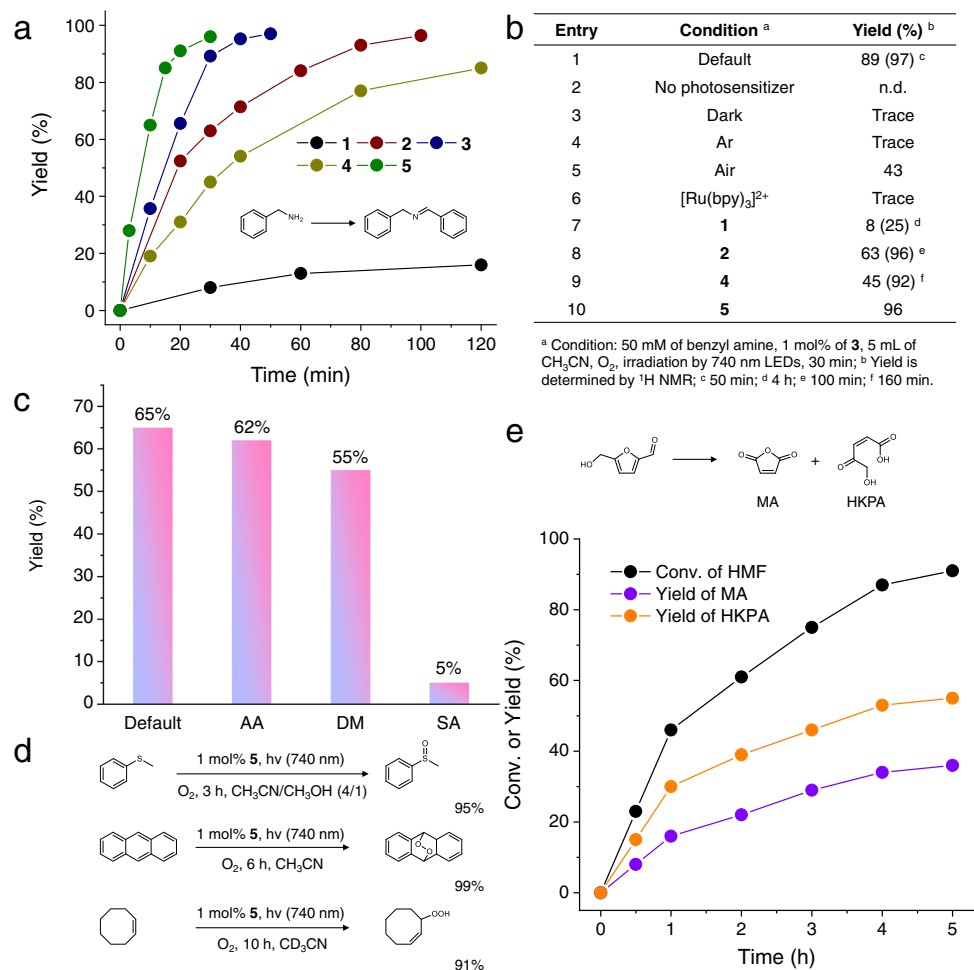

**Fig. 3 Photocatalytic ¹O₂-involved organic reactions using Ru TPA photosensitizers under NIR irradiation.** **a** Yield evolution of benzyl amine homocoupling during photocatalysis using different photocatalysts of **1–5**. Condition: 50 mM benzyl amine, 1 mol% of photocatalyst, 5 mL CH₃CN, O₂ atmosphere, irradiation at 740 nm, room temperature. **b** Comparison of photocatalytic benzyl amine homocoupling performed under various conditions. **c** Photocatalytic yields of benzyl amine homocoupling obtained with or without scavengers. Condition: 50 mM benzyl amine, 50 mM scavenger, 1 mol% **5**, 5 mL CH₃CN, O₂, irradiation at 740 nm, room temperature, 10 min. Default without scavenger, AA L-ascorbic acid, DM D-mannitol, SA sodium azide. **d** Other selected ¹O₂ reactions: thioanisole sulfoxidation, anthracene [4 + 2] Diels-Alder reaction, and cyclooctene oxidation using **5** as the photocatalyst. **e** Photocatalytic HMF oxidation by ¹O₂ to yield maleic acid anhydride (MA) and 5-hydroxy-4-keto-penteroic acid (HKPA) using **5** as the photocatalyst. Condition: 50 mM HMF, 1 mol% **5**, 5 mL CH₃CN, O₂, irradiation at 740 nm, room temperature.

the better performance observed for **5**, this complex was used as the photosensitizer in all the subsequent photocatalytic experiments unless noted otherwise. A non-linear relationship between the yield of benzyl amine C–N coupling product and the LED power was observed, in agreement with the TPA nature of **5** as the photosensitizer (Supplementary Fig. 32)[23,54]. The reaction performance upon one-photon (456 nm) versus two-photon (740 nm) irradiation was also compared in Supplementary Figs. 33–35, in which ten reaction tubes were bundled in one row and the irradiation light was only allowed to excite the reaction tubes from the left side while all the other directions were covered by aluminum foil. It was found that the 456 nm LED was only able to achieve a decent yield (90%) in the 1st reaction tube while all the remaining tubes exhibited no conversion at all, due to the limited penetration of the 456 nm light. In striking contrast, the 740 nm LED was able to not only produce nearly 100% yield in the 1st tube, but also decent yields (91–72%) in the 2nd, 3rd, and 4th tubes. Even the last tube (#10) also produced a yield of 26%. These results unambiguously demonstrate the advantage of deep penetration of NIR light via two-photon absorption of our photosensitizer **5**.

As shown in Fig. 3c, the presence of ascorbic acid and D-mannitol, which are O₂⁻ and OH⁻• scavengers, respectively, did not suppress the formation of N-benzylidenebenzylamine appreciably. However, upon the addition of sodium azide, a ¹O₂ scavenger, the reaction yield decreased from 65% to 5% after 10 min of irradiation at 740 nm. Therefore, these results support the conclusion that ¹O₂ is the primary oxidant for the homocoupling of benzyl amine, formed from the energy transfer process from the excited photosensitizer (e.g., **5**\*) to ³O₂.

Such an extraordinarily high photocatalytic activity of **5** in ¹O₂-involved benzyl amine coupling prompted us to explore other NIR light-driven ¹O₂ reactions, especially because O₂ is regarded as a green oxidant. For example, thioanisole sulfoxidation (Supplementary Figs. 36, 37)[55], anthracene-¹O₂ [4 + 2] Diels-Alder reaction (Supplementary Figs. 38, 39)[56], and allylic hydroperoxide formation from cyclooctene (Supplementary Figs. 40, 41)[57] were smoothly executed with excellent yields (>90%) using **5** as the sole photosensitizer upon 740 nm irradiation under ambient conditions (Fig. 3d). Furthermore, ¹O₂-driven upgrading of biomass-derived platform chemicals,

**Fig. 4 Photocatalytic redox reactions using 5 as the photosensitizer under NIR irradiation. a** Photocatalytic hydrodehalogenation of α-bromoketone using 0.2 mol% **5** and 10 eq. TEOA in CH$_3$CN under 740 nm irradiation for 8 h. **b** Photocatalytic C–H cyanation of tetrahydroisoquinoline using 1 eq. TsCN and 1 mol% **5** in CH$_3$CN under 740 nm irradiation for 24 h. **c** Photocatalytic allylation of benzaldehydes with 10 eq. allylic acetate using 5 mol% **5**, 15 mol% Ni(OTf)$_2$, 20 mol% o-phenanthroline (phen), and 10 eq. DIPEA in CH$_3$CN under 740 nm irradiation for 72 h. All the photocatalytic reactions were performed at room temperature in deaerated solvents under argon.

such as 5-hydroxymethylfurfural (HMF), can also be realized. Electrocatalytic and photocatalytic valorization of HMF has attracted the intense interest of many scientists, including our group. The most common product from HMF oxidation is either 2,5-furandicarboxylic acid[58–60] or 2,5-diformylfuran[61,62], both of which are valuable polymer precursors. In contrast, $^1O_2$-initiated HMF oxidation can result in other value-added fine chemicals, such as maleic acid anhydride (MA) and 5-hydroxy-4-keto-2-pentenoic acid (HKPA), but these reactions remain underexplored[63,64]. Herein, we demonstrate that using **5** as the photosensitizer, NIR light irradiation ($\lambda_{irr}$ = 740 nm) was able to drive the transformation of HMF to MA (55% yield) and HKPA (37% yield) in the presence of O$_2$ with an overall HMF conversion of 92% within 5 h (Fig. 3e and Supplementary Fig. 42). It has been suggested that an ozonide intermediate is formed during the transformation, analogous to the above [4 + 2] Diels-Alder mechanism of anthracene oxidation by $^1O_2$.

**Photocatalytic redox reactions under NIR light irradiation**. In addition to the above $^1O_2$-involved reactions resulting from an energy transfer process, complex **5** is equally effective in driving electron-transfer photoredox reactions upon 740 nm irradiation. Herein, the hydrodehalogenation of phenacyl bromide was selected as a representative reaction for reductive photocatalysis. [Ru(bpy)$_3$]$^{2+}$ has been reported as a competent photocatalyst for this reaction under visible light irradiation[65], however no product was formed upon irradiation at 740 nm. In striking contrast, phenacyl bromide underwent hydrodehalogenation smoothly in the presence of 0.2 mol% of **5** and 10 equivalents of triethanolamine (TEOA) under 740 nm irradiation in deaerated acetonitrile, in which TEOA was employed as the sacrificial electron/proton donor (Fig. 4a). An 86% yield of acetophenone was

obtained after 8 h photocatalysis at room temperature (Supplementary Fig. 43).

In the absence of sacrificial reagents, **5** is also able to drive redox neutral reactions, which utilizes both the reduction and oxidation power of a photocatalyst and thus maximizes the energy efficiency and atom economy. In this case, C-H cyanation of tetrahydroisoquinoline was selected as a representative reaction, wherein p-toluenesulfonyl cyanide (TsCN) was employed as the cyanide source (Fig. 4b). Upon irradiation at 740 nm, the excited state of **5**, **5**$^*$, was oxidatively quenched by TsCN to produce [**Ru**]$^{3+}$ (one-electron oxidized **5**), CN$^-$, and a sulfonyl radical. The resulting [**Ru**]$^{3+}$ complex was able to oxidize tetrahydroisoquinoline to yield an aminium radical cation, resulting in the generation of a neutral carbon radical after α-H deprotonation. Subsequent oxidation of the carbon radical by the sulfonyl radical forms an iminium cation and the final nucleophilic attack of CN$^-$ to the iminium cation furnished the formation of the cyanation product. Overall, two single-electron transfer processes are required in this photocatalytic cycle and a 96% yield is obtained after 24 h irradiation at 740 nm (Supplementary Figs. 44, 45).

In addition to acting as a sole photocatalyst, **5** can also cooperate with transition metal cocatalysts for metallaphotoredox catalysis under NIR light irradiation. As shown in Fig. 4c, in the presence of Ni$^{2+}$, o-phenanthroline, and a sacrificial electron donor DIPEA (diisopropylethylamine), **5** was able to initiate the allylation of aromatic aldehydes with allyl acetate under 740 nm excitation. Good to excellent yields (57–87%) were achieved (Supplementary Figs. 46–48), in which the relatively lower yield of 1-(4-methoxyphenyl)but-3-en-1-ol (57%) was likely due to the lower electrophilicity of the aldehyde group in 4-methoxybenzaldehyde because of the electron donating capability of its methoxy substituent at the para position. Recently, Ru- and Ir-based one-photon-absorbing

photosensitizers have been reported to integrate with Ni[66] and Co[67] cocatalysts for similar reactions, hence it is anticipated that the current NIR light-driven allylation of aldehydes follows an analogous catalytic cycle (Supplementary Fig. 49). Two single-electron-transfer processes between the reduced Ru photocatalyst and Ni species are responsible for the net reductive $C_{sp}^2$-$C_{sp}^3$ coupling transformation.

In summary, five Ru polypyridyl complexes coordinated by ligands with extended π-conjugation were demonstrated to possess excellent photocatalytic performance under NIR light irradiation following a two-photon absorption strategy. Both energy- and electron-transfer reactions were realized upon excitation by low energy 740 nm photons. Activity analysis of these complexes results in the following findings. (i) A greater number of bpyvp-ligands in coordination sphere of the complex generally led to better TPA photocatalysis, in agreement with the increased TPA cross section (e.g., **3** as compared to **1** and **2**). (ii) Electron-donating substituents at the *para* positions of the terminal phenyl groups in the bpyvp-type ligands are beneficial for the overall photocatalytic performance (e.g., **5** as compared to **3** and **4**), likely due to improved intra-molecular charge transfer leading to better two-photon absorption. It is worth noting that **5** also has longer excited lifetime than **3** and **4**. (iii) These TPA Ru complexes can perform in an analogous fashion as one-photon-absorbing photosensitizers, such that a wide range of applications can be envisioned for many homogeneous reactions under NIR light irradiation. Finally, because of better penetration into reaction media and less competing absorption by organic substrates when using NIR light, two-photon-absorbing photo-sensitizers present a new and exciting avenue for homogeneous photocatalysis at large.

## Methods

**Computational methods.** All calculations were performed with the Gaussian 16[68] program package employing the DFT method with Becke's three-parameter hybrid functional and Lee-Yang-Parr's gradient corrected correlation functional (B3LYP)[69–71]. The Stuttgart/Dresden (SDD) basis set and effective core potential were used for the Ru atom[72,73], and 6–31 G* basis set was applied for H, C, O, N and F[74]. The geometries of the singlet ground states of compounds were optimized in $CH_3CN$ using the conductive polarizable continuum model (CPCM). The local minimum on each potential energy surface was confirmed by frequency analysis. Time-dependent DFT calculations produced the singlet excited states of each compound starting from the optimized geometry of the corresponding singlet ground state, using the CPCM method with $CH_3CN$ as the solvent. The calculated absorption spectra, electronic transition contributions, and electron density dif-ference maps (EDDMs) were generated by GaussSum 3.0[75]. The electronic orbitals were visualized using VMD 1.9.4a51[76].

**Materials and instrumentation.** All starting materials were commercially avail-able and used without further purification unless otherwise noted. $^1H$ and $^{13}C\{^1H\}$ and $^{19}F$ NMR spectra were recorded in the designated solvents on a Bruker AV 400 MHz spectrometer. Absorption spectra were measured using a Cary 8454 UV-vis spectrophotometer (Agilent Technologies). Emission spectra were recorded using a Varian Cary Eclipse fluorescence spectrophotometer. Electrochemical measurements were performed on a VMP-3 potentiostat (Biologic Science Instrument) with a three-electrode configuration. Pt wire, Standard Calomel Electrode (SCE), and glassy carbon (GC) were used as the counter, reference, and counter electrode, respectively. All the potentials were calibrated by the redox potential of $Fc^{+/0}$ (Fc: ferrocene). Cyclic voltammetry and square wave voltam-metry experiments were conducted for each complex (**1–5**) in 0.1 M $LiClO_4$ of DMF. Femtosecond transient absorption spectra were performed on a home-built system. The 800 nm (8 mJ 1 kHz rep. rate, ~35 fs pulse width) laser beam is generated from an Astrella laser system then split by the Ti:Saph crystal to form the pump beam (3 mJ) and probe beam. The pump beam goes through an optical parametric amplifier (Coherent OPerA Solo) to generate the desire wavelengths (3-4 uJ). Samples are loaded into a flow cell opened to air. All samples were at a concentration that afforded an absorption of ~0.6 − 0.8 at the sample excitation wavelength. Fluorescence emission decay was measured on a LP980 spectrometer system (Edinburgh Instruments). The 355 nm laser beam is generated by a frequency-tripled Quanta-Ray INDI Nd:YAG laser (Spectra-Physics, ~6 ns pulses at 10 Hz) then goes through a tunable optical parametric oscillator (Spectra-Phy-sics) to output the desired wavelengths, and the white light probe beam is generated from a 150 W Xe arc lamp. Single wavelength emission decay traces were collected using a PMT and digital oscilloscope (Tektronix MDO3022, 200 MHz, 2.5 GS/s).

Spectral measurements were collected using an iCCD camera (iStar, Andor Technology). All the air-tight solutions were purged with $N_2$ for 15 min before experiments. The absorbance of the solutions was kept $0.6 − 0.8$ at the excited wavelength. The absorption spectra were taken before and after the experiments to confirm no degradation over the measurements.

## Synthesis

*(E,E')-4,4'-Bisstyryl]-2,2'-bipyridine (bpyvp-H).* bpyvp-H was prepared according to the reported literature with slight modification[50]. A solution of 4,4'-dimethyl-2,2'-bipyridine (0.92 g, 5 mmol) and $K^tOBu$ (2.3 g, 20 mmol) in dry DMF (50 mL) was stirred for 1 h under Ar. Benzaldehyde (1524.7 μL, 15 mmol) was then added to the reaction mixture. After stirring at room temperature for 24 h, the solution was treated with 400 mL water and the suspension was stored at 5 °C for several hours. The precipitated solid was filtered and washed with methanol via Soxhlet extraction for 24 h and dried in vacuum at 60 °C to afford 1.49 g light yellow solid product (yield: 83%). $^1H$ NMR (400 MHz, $CDCl_3$): δ 8.61 (d, $J = 5.1$ Hz, 1H), 8.49 (d, $J = 1.7$ Hz, 1H), 7.51 (d, $J = 7.2$ Hz, 2H), 7.45–7.22 (m, 5H), 7.08 (d, $J = 16.3$ Hz, 1H). $^{13}C\{^1H\}$ NMR (101 MHz, $CDCl_3$): 156.7, 149.8, 146.0, 136.5, 133.6, 129.1, 128.9, 127.3, 126.4, 121.3, 118.5.

*(E,E')-4,4'-Bis[p-fluorostyryl]-2,2'-bipyridine (bpyvp-F).* bpyvp-F was synthesized following a similar procedure as the above bpyvp-H. A solution of 4,4'-dimethyl-2,2'-bipyridine (0.92 g, 5 mmol and $K^tOBu$ (2.3 g, 20 mmol) in dry DMF (50 mL) was stirred for 1 h under Ar. *p*-Fluorobenzaldehyde (1609.2 μL, 15 mmol) was then added to the reaction mixture. After stirring at room temperature for 24 h, the solution was treated with 400 mL water and the suspension was stored at 5 °C for several hours. The precipitated solid was filtered and washed with methanol via Soxhlet extraction for 24 h and dried in vacuum at 60 °C to afford 1.78 g light yellow solid product (yield: 90%). $^1H$ NMR (400 MHz, $CDCl_3$): δ 8.68 (d, $J = 5.1$ Hz, 1H), 8.55 (s, 1H), 7.55 (dd, $J = 8.6$, 5.4 Hz, 2H), 7.47–7.36 (m, 2H), 7.14–7.03 (m, 3H). $^{13}C\{^1H\}$ NMR (101 MHz, $CDCl_3$):164.4, 161.9, 156.7, 149.8, 145.8, 132.7, 132.4, 128.9, 128.8, 126.11, 121.3, 118.4, 116.2, 116.0. $^{19}F$ NMR (376 MHz, $CDCl_3$): -112.3.

*(E,E')-4,4'-Bis(p-methoxystyryl)-2,2'-bipyridine (bpyvp-OMe).* bpyvp-OMe was synthesized following a similar procedure as the above bpyvp-H. A solution of 4,4'-dimethyl-2,2'-bipyridine (0.92 g, 5 mmol) and $K^tOBu$ (2.3 g, 20 mmol) in dry DMF (50 mL) was stirred for 1 h under Ar. *p*-Methoxybenzaldehyde (1823.4 μL, 15 mmol) was then added to the reaction mixture. After stirring at room tem-perature for 24 h, the solution was treated with 400 mL water and the suspension was stored at 5 °C for several hours. The precipitated solid was filtered and washed with methanol via Soxhlet extraction for 24 h and dried in vacuum at 60 °C to afford 1.68 g light yellow solid product (yield: 80%). $^1H$ NMR (400 MHz, $CDCl_3$): δ 8.65 (d, $J = 5.0$ Hz, 1H), 8.52 (s, 1H), 7.55–7.34 (m, 4H), 7.05–6.90 (m, 3H), 3.85 (s, 3H). $^{13}C\{^1H\}$ NMR (101 MHz, $CDCl_3$): 160.4, 156.7, 149.7, 146.3, 133.1, 129.3, 128.6, 124.2, 121.1, 118.3, 114.5, 55.6.

*[Ru(2,2'-bipyridine)₂((E,E')-4,4'-Bisstyryl]-2,2'-bipyridine)][PF₆]₂ (1).* **1** was pre-pared according to the reported literature with slight modification[10]. A mixture of $Ru(bpy)_2Cl_2$ (0.2 mmol, 0.097 g) and bpyvp-H (0.3 mmol, 0.108 g) was suspended in $DMF/H_2O$ (v/v, 30/30 mL). The mixture was then refluxed at 100 °C for 8 h under argon atmosphere. After cooling to room temperature, the solvent was removed under reduced pressure. The residue was dissolved in methanol and added with a saturated aqueous solution of $NH_4PF_6$. The red precipitated solid product (**1**) was filtered and washed with water and diethyl ether with a yield of 46% (98 mg). $^1H$ NMR (400 MHz, $CD_3CN$): δ 8.73 (s, 2H), 8.51 (d, $J = 8.2$ Hz, 4H), 8.06 (t, $J = 7.9$ Hz, 4H), 7.83–7.60 (m, 12H), 7.50–7.33 (m, 12H), 7.32 (d, $J = 16.3$ Hz, 2H). $^{13}C\{^1H\}$ NMR (101 MHz, $CD_3CN$): 158.1, 157.9, 152.6, 152.4, 147.7, 138.7, 137.4, 136.7, 130.6, 130.1, 128.5, 128.4, 125.3, 125.2, 125.0, 121.6.

*[Ru(2,2'-bipyridine)((E,E')-4,4'-Bisstyryl]-2,2'-bipyridine)₂][PF₆]₂ (2).* **2** was pre-pared according to a similar procedure of that of **1**. A mixture of bpyvp-H (0.5 mmol, 0.18 g), $Ru(DMSO)_4Cl_2$ (0.25 mmol, 0.12 g) and lithium chloride (25 mmol, 1.06 g) in dry DMF (30 mL) was refluxed for 5 h under argon atmo-sphere. The solution was cooled down to room temperature and water (200 mL) was added. The solid precipitate was filtrated and washed with water and diethyl ether to give the intermediate compound of [Ru((E,E')-4,4'-bisstyryl-2,2'-bipyridine)₂Cl₂] without further purification. [Ru((E,E')-4,4'-Bisstyryl-2,2'-bipyr-idine)₂Cl₂] and 2,2'-bipyridine (0.2 mmol, 0.18 g) were suspended in $DMF/H_2O$ (v/v, 30/30 mL). The mixture was then refluxed at 100 °C for 8 h under argon atmosphere. After cooling to room temperature, the solvent was removed under reduced pressure. The residue was dissolved in methanol and added with a saturated aqueous solution of $NH_4PF_6$. The dark-red precipitated solid product (**2**) was filtered and washed with water and diethyl ether with a yield of 67% (170 mg). $^1H$ NMR (400 MHz, $CD_3CN$): δ 8.75 (s, 4H), 8.52 (d, $J = 7.8$ Hz, 2H), 8.07 (t, $J = 7.8$ Hz, 2H), 7.60–7.89 (m, 18H), 7.53–7.41 (m, 18H), 7.33 (d, $J = 16.4$ Hz, 4H). $^{13}C\{^1H\}$ NMR (101 MHz, $CD_3CN$): 158.2, 157.9, 152.6, 152.4, 147.6, 138.7, 137.3, 136.8, 130.6, 130.1, 128.5, 128.4, 125.3, 125.2, 125.0, 121.6.

*[Ru((E,E')-4,4'-Bisstyryl)]-2,2'-bipyridine)₃][PF₆]₂ (3).* Ru(DMSO)₄Cl₂ (0.25 mmol, 0.12 g, 1 equiv.) and bpyvp-H (1 mmol, 0.36 g, 4 equiv.) were refluxed in dry ethanol (150 mL) under argon atmosphere for 15 h. After cooling down to the room temperature, the residue solid was filtered off. And the remaining solution was added with a saturated aqueous solution of NH₄PF₆. The dark-red precipitated solid product (3) was filtered and washed with water and diethyl ether with a yield of 80% (294 mg). $^1$H NMR (400 MHz, CD₃CN): δ 8.76 (s, 6H), 7.82–7.74 (m, 12H), 7.70 (d, $J$ = 7.5 Hz, 12H), 7.54–7.4 (m, 24H), 7.33 (d, $J$ = 16.3 Hz, 6H). $^{13}$C{$^1$H} NMR (101 MHz, CD₃CN): 158.2, 152.4, 147.6, 137.3, 136.8, 130.6, 130.1, 128.4, 125.3, 125.1, 121.6.

*[Ru((E,E')-4,4'-(p-fluorostyryl))]-2,2'-bipyridine)₃][PF₆]₂ (4).* 4 was prepared via a similar procedure of that of 3. Ru(DMSO)₄Cl₂ (0.25 mmol, 0.12 g, 1 equiv.) and bpyvp-F (1 mmol, 0.36 g, 4 equiv.) were refluxed in dry ethanol (150 mL) under argon atmosphere for 15 h. After cooling down to the room temperature, the residue solid was filtered off. And the remaining solution was added with a saturated aqueous solution of NH₄PF₆. The dark-red precipitated solid product (4) was filtered and washed with water and diethyl ether with a yield of 75% (296 mg). $^1$H NMR (400 MHz, CD₃COCD₃): δ 9.04 (s, 6H), 8.12–7.98 (m, 6H), 7.88–7.63 (m, 24H), 7.41 (d, $J$ = 16.4 Hz, 6H), 7.27–7.05 (m, 12H). $^{13}$C{$^1$H} NMR (101 MHz, CD₃CN): 163.0, 158.1, 152.4, 136.0, 133.2 (d, $J$ = 3.5 Hz), 130.4 (d, $J$ = 8.4 Hz), 125.1 (d, $J$ = 34.5 Hz), 121.5, 117.1, 116.8. $^{19}$F NMR (376 MHz, CD₃CN): -72.6 (d, $J$ = 704.6 Hz), -112.8.

*[Ru((E,E')-4,4'-(p-methoxystyryl))]-2,2'-bipyridine)₃][PF₆]₂ (5).* 5 was prepared via a similar procedure of that of 3. Ru(DMSO)₄Cl₂ (0.25 mmol, 0.12 g, 1 equiv.) and bpyvp-F (1 mmol, 0.36 g, 4 equiv.) were refluxed in dry ethanol (150 mL) under argon atmosphere for 15 h. After cooling down to the room temperature, the residue solid was filtered off. And the remaining solution was added with a saturated aqueous solution of NH₄PF₆. A saturated aqueous solution of NH₄PF₆ was added. The dark-red precipitated solid product (5) was filtered and washed with water and diethyl ether with a yield of 73% (303 mg). $^1$H NMR (400 MHz, CD₃CN): δ 8.68 (s, 6H), 7.71 (m, 12H), 7.64 (d, $J$ = 8.5 Hz, 12H), 7.45 (d, $J$ = 5.8 Hz, 6H), 7.17 (d, $J$ = 16.3 Hz, 6H), 7.02 (d, $J$ = 8.3 Hz, 12H), 3.84 (s, 18H). $^{13}$C{$^1$H} NMR (101 MHz, CD₃CN): 162.0, 158.1, 152.1, 147.9, 136.9, 130.0, 129.4, 124.9, 122.6, 121.2, 118.3, 115.5, 56.1.

## Photocatalysis

*Benzyl amine C-N coupling.* In a typical experiment, a solution of 50 mM benzyl amine and 1 mol% photosensitizer in 5 mL CH₃CN was added in a sealed 20 mL vial. After bubbling with O₂ for 10 min, the solution was irradiated under near-IR LED (PR160L-740-C, 8.18 W) for a certain period of time to obtain N-benzylidenebenzylamine as the product. $^1$H NMR (400 MHz, CDCl₃): δ 8.32 (s, 1H), 7.70 (d, $J$ = 8.9 Hz, 2H), 7.17–7.35 (m, 8H), 4.75 (s, 2H). For scavenger control experiments, the concentration of scavenger was 50 mM, photosensitizer was complex 5 and the reaction time was 10 min. For power dependence experiment, the light source was 730 nm LED (M730L5, Thorlabs) and a 695 nm long-pass filter (SCHOTT RG695) was placed between the LED and the reaction vessel. The LED power was controlled via the LED driver (DC2200, Thorlabs).

*Thioanisole sulfoxidation.* In a typical procedure, a solution of 50 mM thioanisole and 1 mol% 5 in 4 mL CH₃CN and 1 mL methanol was added in a sealed 20 mL vial. After bubbling with O₂ for 10 min, the solution was irradiated under near-IR LED (PR160L-740-C, 8.18 W) for a certain period of time to obtain methyl phenyl sulfoxide as the product (yield: 95%). $^1$H NMR (400 MHz, CDCl₃): δ 7.71–7.40 (m, 5H) 2.69 (s, 1H).

*Anthracene-O₂ [4 + 2] Diels-Alder reaction.* In a typical procedure, a solution of 10 mM anthracene and 1 mol% 5 in 5 mL CH₃CN was added in a sealed 20 mL vial. After bubbling with O₂ for 10 min, the solution was irradiated under near-IR LED (PR160L-740-C, 8.18 W) for a certain period of time to obtain 9,10-dihydro-9,10-epidioxyanthracene as the product (yield: 99%). $^1$H NMR (400 MHz, CDCl₃): δ 7.56–7.16 (m, 8H) 6.04 (s, 2H).

*Cyclooctene ene-type reaction.* In a typical procedure, a solution of 50 mM anthracene and 1 mol% 5 in 5 mL CD₃CN was added in a sealed 20 mL vial. After bubbling with O₂ for 10 min, the solution was irradiated under near-IR LED (PR160L-740-C, 8.18 W) for a certain period of time to obtain 3-hydroperoxycyclooct-1-ene as the final product (yield: 91%). $^1$H NMR (400 MHz, CDCl₃): δ 9.39 (brs, 1H), 5.75–5.61 (m, 1H), 5.60–5.53 (m, 1H), 4.85–4.69 (m, 1H), 2.29–1.83 (m, 4H), 1.75–1.17 (m, 6H).

*HMF upgrading.* In a typical procedure, a solution of 50 mM HMF and 1 mol% 5 in 5 mL CH₃CN was added in a sealed 20 mL vial. After bubbling with O₂ for 10 min, the solution was irradiated under near-IR LED (PR160L-740-C, 8.18 W) for a certain period of time to obtain maleic acid anhydride (yield: 55%) and 5-hydroxy-4-keto-penteroic acid (yield: 37%) as the products. Maleic acid anhydride: $^1$H NMR (400 MHz, CDCl₃): δ 7.04 (s, 1H). 5-Hydroxy-4-keto-penteroic acid: $^1$H NMR (400 MHz, CDCl₃): δ 6.94 (d, $J$ = 10.3 Hz, 1H), 6.84 (d, $J$ = 10.3 Hz, 1H), 4.99 (s, 1H).

*Phenacyl bromide dehalogenation.* In a typical procedure, a solution of 50 mM phenacyl bromide, 10 equiv. triethanolamine (TEOA), and 0.2 mol% 5 in 5 mL CH₃CN was added in a sealed 20 mL vial. After bubbling with Ar for 10 min, the solution was irradiated under near-IR LED (PR160L-740-C, 8.18 W) for 8 h. The product was purified by silica gel column chromatography with $V_{hexanes}/V_{ethyl\ acetate}$ = 100/3 to afford acetophenone as the product (yield: 86%). $^1$H NMR (400 MHz, CDCl₃): δ 8.04 – 7.88 (m, 2H), 7.62 – 7.52 (m, 1H), 7.46 (dd, $J$ = 8.3, 7.0 Hz, 2H), 2.60 (s, 3H).

*Cyanation of tetrahydroisoquinoline.* The starting substrate 2-(4-methoxyphenyl)-1,2,3,4-tetrahydroisoquinoline was synthesized based on a published procedure[77]. A mixture of 1,2,3,4-tetrahydroisoquinoline (0.63 mL, 5 mmol) and 4-iodoanisole (1.2 g, 5 mmol), copper(I) iodide (95.3 mg, 0.5 mmol), and potassium phosphate (2.1 g, 10 mmol) in 2-propanol/ethylene glycol (20/2, v/v) was refluxed at 80 °C for 24 h under argon atmosphere. After cooling down to room temperature, the mixture was quenched by water (50 mL) and extracted with ethyl acetate (30 mL × 3). The organic layer was concentrated under reduced pressure via rotary evaporation. The obtained residue was purified by silica gel column chromatography to afford the white solid product (860 mg, 72%). $^1$H NMR (CDCl₃, 400 MHz): δ 7.19–7.25 (m, overlapped, 4H), 7.05 (d, $J$ = 9.0 Hz, 2H), 6.96 (d, $J$ = 9.0 Hz, 2H), 4.37 (s, 2H), 3.84 (s, 3H), 3.50 (t, $J$ = 5.8 Hz, 2H), 3.04 (t, $J$ = 5.7 Hz, 2H).

The following photocatalytic cyanation process was conducted in a sealed 20 mL vial containing a solution of 50 mM 2-(4-methoxyphenyl)-1,2,3,4-tetrahydroisoquinoline, 50 mM p-toluenesulfonyl cyanide, and 1 mol% 5 in 5 mL CH₃CN. After bubbling with Ar for 10 min, the solution was irradiated under near-IR LED (PR160L-740-C, 8.18 W) for 24 h. The product was purified by silica gel column chromatography with $V_{hexanes}/V_{ethyl\ acetate}$ = 100/3 to afford 2-(4-methoxyphenyl)-1,2,3,4-tetrahydroisoquinoline-1-carbonitrile as the product (yield: 96%). $^1$H NMR (400 MHz, CDCl₃): δ $^1$H NMR (400 MHz, CDCl3) δ 7.41–7.20 (m, 4H), 7.13–7.05 (m, 2H), 6.98–6.88 (m, 2H), 5.37 (s, 1H), 3.81 (s, 3H), 3.64–3.54 (m, 1H), 3.44 (m, 1H), 3.17 (m, 1H), 2.94 (m, 1H).

*Ni-assisted $C_{sp}^2$–$C_{sp}^3$ coupling.* In a typical procedure, a solution of 50 mM para-substituted benzaldehyde (-H, -OCH₃, or -CF₃), 10 equiv. allyl acetate, 10 equiv. N,N-diisopropylethylamine (DIPEA), 5 mol% 5, 15 mol% Ni(OTf)₂, and 20 mol% o-phenanthroline in 2 mL CH₃CN was added in a sealed 5 mL vial. After bubbling with Ar for 3 min, the solution was irradiated under near-IR LED (PR160L-740-C, 8.18 W) for 72 h. The product was purified by silica gel column chromatography with $V_{hexanes}/V_{ethyl\ acetate}$ = 10/1 to afford para-substituted 1-phenyl-3-buten-1-ol as the product (yield: 57 – 87%). 1-phenyl-3-buten-1-ol: $^1$H NMR (400 MHz, CDCl₃): δ 7.41–7.22 (m, 5H), 5.81 (m, 1H), 5.22–5.09 (m, 2H), 4.73 (m, 1H), 2.60–2.43 (m, 2H), 2.22–2.17 (brs, 1H). $^{13}$C{$^1$H} NMR (101 MHz, CDCl₃): δ 144.1, 134.7, 128.6, 127.7, 126.0, 118.6, 73.5, 44.0. 1-(4-methoxyphenyl)-3-buten-1-ol: $^1$H NMR (400 MHz, CDCl₃): δ 7.31–7.24 (m, 2H), 6.92–6.84 (m, 2H), 5.80 (m, 1H), 5.17 (m, 1H), 5.15–5.08 (m, 1H), 4.69 (t, $J$ = 6.5 Hz, 1H), 3.80 (s, 3H), 2.54–2.46 (m, 2H). $^{13}$C{$^1$H} NMR (101 MHz, CDCl₃): δ 159.2, 136.3, 134.8, 127.3, 118.3, 114.0, 73.2, 55.5, 43.9. 1-(4-(trifluoromethyl)phenyl)-3-buten-1-ol: $^1$H NMR (400 MHz, CDCl₃): δ 7.61 (d, $J$ = 8.1 Hz, 1H), 7.47 (d, $J$ = 8.1 Hz, 1H), 7.37–7.23 (m, 2H), 5.86–5.70 (m, 1H), 5.23–5.12 (m, 2H), 4.76 (m, 1H), 2.60–2.38 (m, 2H). $^{13}$C{$^1$H} NMR (101 MHz, CDCl₃): δ 148.0, 142.5, 134.2, 133.9, 128.8, 127.4, 126.3, 119.4, 119.1, 72.7, 44.1.

## Data availability

Characterization and calculation of compounds can be found in the Supplementary Information.

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

## Acknowledgements

Y.S. acknowledges the financial support of the National Science Foundation (CHE-1955358), University of Cincinnati, and the Ohio Supercomputer Center. C.T. is grateful for grants from the National Science Foundation (CHE-2102508) and Department of Energy (DE-SC002043) for partial support of this work.

## Author contributions

Y.S. conceived and designed this research. G.H. synthesized and characterized the photosensitizers and conducted the theoretical computation. G.H., G.L. and C.H. performed photocatalysis experiments. J.H. and C.T. carried out the transient absorption and lifetime measurements. All authors contributed to the analysis and interpretation of results. Y.S., G.H., and C.T. wrote the manuscript.

## Competing interests

The authors declare no competing interests.
