## [Peer Review File · Nature Communications]

REVIEWER COMMENTS

Reviewer #1 (Remarks to the Author):

Turro, Sun and coworkers report on novel photocatalytic applications driven by commercial NIR LEDs.

Inspired by recent studies on PDT sensitizers with high two-photon cross-sections (e.g. Ref. 32, which

already deals with "complex 5" used by Turro and Sun as well) the authors prepared, investigated and

exploited some Ru complexes with extended pi-system for several different photocatalytic applications.

The test reactions are typically carried out with blue light under conventional one-photon conditions,

but the NIR excitation approach that they present provides inherent advantages such as deep penetration depth

and improved selectivity.

The authors combined several experimental as well as theoretical techniques including time-resolved optical spectroscopy with different excitation wavelengths, DFT calculations and many irradiation experiments

to understand the important findings.

In contradiction to what can be found in physical chemistry or photochemistry textbooks ("pulsed lasers are required for

the simultaneous absorption of two photons"), the authors postulate that this kind of chemistry can be carried out

with NIR LEDs. This can be regarded as breakthrough in homogeneous photocatalysis and the paper would

certainly trigger the further development of both NIR-driven catalysis and the simultaneous two-photon absorption.

However, prior to publication of this manuscript in Nature Comm., several mechanistic control experiments

have to be carried out to fully support their claims as detailed below.

Major issue:

1) The 800 nm laser experiments presented in Fig. 2 clearly demonstrate that a two-photon excitation is going on.

However, calculating the laser power and comparing it to the LED power (typically 100 to 200 mW/cm²) one finds

that the laser intensity is higher by about ten orders of magnitude. Even with an outstanding two-photon cross-section,

the simultaneous two-photon absorption can be regarded as very unlikely. That reasoning is also borne out by

a recent review on two-photon absorption processes with useful calculations and predictions (Kobayashi et al., J. Photochem. Photobiol. C, 2018)

and by a review article in Angew. Chem. Int. Ed. (Wenger and co-workers, 2020), in which consecutive two-photon reactions with

singlet- and triplet-excited states are shown to require pulsed lasers or a focused cw laser.

From the LED number in the Supplementary Materials part of the paper, I could extract the piece of information that

the 740 nm LED was prepared by Kessil and luckily, I could find the LED output spectrum as we recently contacted Kessil for a

quotation. The 740 nm LED has an emission onset at about 640 nm (I tried to attach a figure of

the LED emission but I am not sure if this will work). The molar absorption coefficients of the catalysts are clearly non-zero at about 650 nm and they used catalyst concentrations of up to 2.5 mM during photocatalysis. To put a long story short, I have the impression that there is a certain overlap of LED emission and catalyst absorption bands such that conventional one-photon excitation is going on (in line with the observation that catalyst 5 shows the best performance)

The authors could prove me wrong by showing that the reactions work with identical performance when 700 nm long-pass filters (sometimes also called cut-off filters) are between LED and reaction flask and I would like to see the power dependence of product formation, as suggested in the Wenger review as easy-to-carry-out experiment. With these successful experiments, I would strongly support the publication of this manuscript.

Minor comments:

2) I have the impression that only a handful of papers dealing with two-photon catalysis have been cited (refs. 4,6 and 9), although two-photon photoredox catalysis is a very hot topic with many recent papers in JACS, Chem. Sci, Angewandte (by e.g. Francis, Wagenknecht, Wickens, Perez-Ruiz, Gang Han, Goez, Giedyk ...). The authors could compare their novel approach to existing methods, thereby highlighting the unique advantages of simultaneous two-photon absorption.

Right now the paper looks rather like a PDT or a conventional one-photon catalysis paper when having a look at the cited literature.

3) Regarding the DFT calculations presented in the main part of the paper: Rounding the predicted energies up/down (i.e. 551 nm instead of 550.7 nm) might be better to avoid an overestimation of the accuracy of these methods.

4) Page 7: I am not selling Kessil lamps but Kessil LEDs are usually considered as high-power LEDs and not low-power LEDs, given that most commercial LEDs have a much lower output power.

5) page 2, "competing absorption by reactants": Perhaps the authors could add side and (catalyst-derived) decomposition products because these species generated in the course of the reaction absorb light as well (to present further advantages of NIR excitation).

6) There is a typo on the first page of the Supplementary Materials document "Supplemenatry Materials".

Reviewer #2 (Remarks to the Author):

In this work, Y. Sun et al. report a group of ruthenium polypyridyl complexes possessing two-photon absorption (TPA) capability for a variety of photocatalytic organic transformations upon irradiation with NIR light. The authors demonstrate that these two-photon absorption (TPA) photosensitizers can operate for both energy- and electron-transfer reactions. These reactions catalyzed by TPA photosensitizers were systematically studied and the results demonstrate that similar reactivity and selectivity compared to those regular photocatalytic reactions. These observations are not surprising since the same excited states of the photosensitizers were

produced by two-photon or one photon excitation. Although authors highlighted TPA photosensitizers present deep penetration into reaction media and negligible competing absorption by organic substrates, this is not the main challenge of photocatalytic organic reactions. Such issue could be easily solved by using flow photoreactor. The utilization of two-photon excitation increases cost and energy consumption. TPA photosensitization is not practically applicable strategy for photocatalytic reactions. It doesn't show clear advantages over the regular photocatalytic reactions. If this type of photosensitizer can be used in biological systems, it might be meaningful. Therefore, the present manuscript doesn't meet the requirement of Nat. Commun. Other comments:

- 1) The molecular structures of Ru complexes 1, 2 and 3 should also be added in figure 1.
- 2) The authors described that "the utilization of UV/visible photons presents several intrinsic limitations such as poor penetration through reaction media, competing absorption by reactants, and incompatibility against substrates bearing light-sensitive functionalities, as well as limited coverage of the solar spectrum." Please cite appropriate reference to support your statement.
- 3) Some photocatalytic organic reactions were used to illustrate that TPA photosensitizers can effectively catalyze organic reactions upon irradiation with 740 nm light. The author ought to compare the photocatalytic conversion efficiency of the reaction upon irradiation with single-photon light source.

Reviewer #3 (Remarks to the Author):

This manuscript by Han et al. describes two-photon-absorbing ruthenium polypyridyl complexes and shows that they can promote a variety of photocatalytic organic reactions involving energy transfer to O₂ or redox transformations of the substrate. The work is innovative, well-described, and written in a concise and accessible fashion. The strategy of using two-photon absorption in organic photocatalysis has the potential to take off, provided suitable photosensitizers with large TPA cross sections can be identified, and this paper shows a way to do that with the widely used Ru-bpy catalyst system. I can identify no major technical flaws with the work. I commend the authors for putting together a nice paper, and I recommend publication without extensive revision.

I only have a few minor suggestions for the authors, which are listed below.

1. The authors should cite this recent work that also uses a two-photon strategy for photoredox catalysis: DOI: 10.1021/jacs.1c07617. This paper came out after the authors submitted this present manuscript for consideration, so I don't blame them for not including it and I don't think it detracts from the novelty and impact of their work in a significant way.
2. In the experimental section and in the figures showing NMR spectra, ¹³C should be written as ¹³C{¹H} in all cases.
3. The ¹³C{¹H} and ¹⁹F NMR data is not reported correctly in the experimental section for complex 4. In the ¹³C{¹H} spectrum there should be one or more peaks reported as doublets due to coupling to ¹⁹F; in particular the carbon bound directly to fluorine should have a large coupling constant. In the ¹⁹F spectrum, the peak for PF₆⁻ should be reported as a doublet with coupling to ³¹P, not as two separate peaks.

Point-by-Point Response Letter

Reviewer 1

Turro, Sun and coworkers report on novel photocatalytic applications driven by commercial NIR LEDs. Inspired by recent studies on PDT sensitizers with high two-photon cross-sections (*e.g.* Ref. 32, which already deals with "complex 5" used by Turro and Sun as well) the authors prepared, investigated and exploited some Ru complexes with extended pi-system for several different photocatalytic applications. The test reactions are typically carried out with blue light under conventional one-photon conditions, but the NIR excitation approach that they present provides inherent advantages such as deep penetration depth and improved selectivity.

The authors combined several experimental as well as theoretical techniques including time-resolved optical spectroscopy with different excitation wavelengths, DFT calculations and many irradiation experiments to understand the important findings. In contradiction to what can be found in physical chemistry or photochemistry textbooks ("pulsed lasers are required for the simultaneous absorption of two photons"), the authors postulate that this kind of chemistry can be carried out with NIR LEDs. This can be regarded as breakthrough in homogeneous photocatalysis, and the paper would certainly trigger the further development of both NIR-driven catalysis and the simultaneous two-photon absorption. However, prior to publication of this manuscript in Nature Comm., several mechanistic control experiments have to be carried out to fully support their claims as detailed below.

Major issue:

1) The 800 nm laser experiments presented in Fig. 2 clearly demonstrate that a two-photon excitation is going on. However, calculating the laser power and comparing it to the LED power (typically 100 to 200 mW/cm²) one finds that the laser intensity is higher by about ten orders of magnitude. Even with an outstanding two-photon cross-section, the simultaneous two-photon absorption can be regarded as very unlikely. That reasoning is also borne out by a recent review on two-photon absorption processes with useful calculations and predictions (Kobayashi et al., J. Photochem. Photobiol. C, 2018) and by a review article in Angew. Chem. Int. Ed. (Wenger and co-workers, 2020), in which consecutive two-photon reactions with singlet- and triplet-excited states are shown to require pulsed lasers or a focused cw laser.

From the LED number in the Supplementary Materials part of the paper, I could extract the piece of information that the 740 nm LED was prepared by Kessil and luckily, I could find the LED output spectrum as we recently contacted Kessil for a quotation. The 740 nm LED has an emission onset at about 640 nm (I tried to attach a figure of the LED emission, but I am not sure if this will work). The molar absorption coefficients of the catalysts are clearly non-zero at about 650 nm and they used catalyst concentrations of up to 2.5 mM during photocatalysis. To put a long story short, I have the impression that there is a certain overlap of LED emission and catalyst absorption bands such that conventional one-photon excitation is going on (in line with the observation that catalyst 5 shows the best performance). The authors could prove me wrong by showing that the reactions work with identical performance when 700 nm long-pass filters (sometimes also called cut-off filters) are between LED and reaction flask and I would like to see the power dependence of product formation, as suggested in the Wenger review as easy-to-carry-out experiment. With these successful experiments, I would strongly support the publication of this manuscript.

Our Response: Following the reviewer's suggestions, we performed two control experiments: (1) photocatalysis with low-power LED and a long-pass filter and (2) power dependence of the NIR light irradiation on reaction yield. The testing reaction is the photocatalytic C-N coupling of benzyl amine.

1. We purchased a new 730 nm LED (Thorlabs, M730L5, 680 mW), in order to alleviate any concerns regarding the high power of those Kessil LEDs, and a 695 nm long-pass filter (SCHOTT RG695) for this control experiment. The new LED emission spectrum and long-pass filter profile are shown in **Figures R1a**

and **R1b**, respectively. It is apparent that any light before 670 nm will be effectively blocked by the 695 nm long-pass filter as the transmittance is less than 0.01 at 670 nm while our complex **5** does not have any absorption beyond 670 nm (see **Figure 2b** in the revised manuscript). Furthermore, it should be noted that the 695 nm long-pass filter does not have 100% transmittance for light beyond 695 nm. The transmittance is about 0.66 at 700 nm, 0.85 at 710 nm, and 0.93-0.98 at longer wavelength. With these considerations in mind, based on the results shown in **Figures R2-R4**, we can conclude that complex **5** is indeed a TPA photocatalyst and the slightly lower yields achieved with the employment of the 695 nm long-pass filter were primarily due to the lower intensity of passed light through the filter.

Figure R1. (a) M730L5 spectrum and LED parameters (<https://www.thorlabs.com/thorproduct.cfm?partnumber=M730L5>). (b) Long-pass filter transmittance spectrum (<https://www.edmundoptics.com/p/rg-695-254mm-dia-longpass-filter/2575/>).

Figure R2. Yield change over time for the photocatalytic C-N coupling of benzyl amine with or without a 695 nm long-pass filter. Condition: 50 mM benzyl amine, 1 mol% of **5**, 2 mL CH₃CN, O₂ atmosphere, irradiation by 730 nm LED (M730L5, 680 mW) at room temperature, with or without a 695 nm long-pass filter (SCHOTT RG695).

Figure R3. ^1H NMR spectra of the photocatalytic C-N coupling of benzyl amine with a 695 nm long-pass filter. Condition: 50 mM benzyl amine, 1 mol% of **5**, 2 mL CH_3CN , O_2 atmosphere, irradiation by 730 nm LED (M730L5, 680 mW) at room temperature with a 695 nm long-pass filter (SCHOTT RG695).

Figure R4. ^1H NMR spectra of the photocatalytic C-N coupling of benzyl amine without a 695 nm long-pass filter. Condition: 50 mM benzyl amine, 1 mol% of **5**, 2 mL CH_3CN , O_2 atmosphere, irradiation by 730 nm LED (M730L5, 680 mW) at room temperature without a 695 nm long-pass filter (SCHOTT RG695).

2. The LED illumination power of our newly purchased 730 nm LED can be finely tuned by the LED driver (Thorlabs, DC2200). Therefore, we performed the same photocatalytic C-N coupling reaction of benzyl amine under 730 nm irradiation of different power. As shown in **Figure R5a**, during the first 20 min photocatalysis, the product yield is linearly increased with the irradiation duration as the starting material concentration is not depleted to become a limiting factor. Consequently, we plotted the yield obtained at 20 min versus the LED power in **Figure R5b** (similar analysis has been reported, please see *Angew. Chem. Int. Ed.* 2020, 59, 10266-10284 and *ACS Catal.* 2019, 9, 422-430). It is apparent that when we increased the LED power from 6.8 to 272 mW, a non-linear increase in yield was observed, in agreement with the two-photon absorption mechanism of our photocatalyst (complex **5** in this case).

Figure R5. (a) Yield change over time upon irradiation at 695 nm of different power. (b) The obtained yields after 20 min irradiation at 695 nm of different power. Conditions: 50 mM benzyl amine, 1 mol% **5**, 2 mL CH₃CN, O₂, irradiation using 730 nm LED (M730L5) of controlled power at room temperature with a 695 nm long-pass filter.

Minor comments:

2) I have the impression that only a handful of papers dealing with two-photon catalysis have been cited (refs. 4,6 and 9), although two-photon photoredox catalysis is a very hot topic with many recent papers in *JACS*, *Chem. Sci*, *Angewandte* (by *e.g.*, Francis, Wagenknecht, Wickens, Perez-Ruiz, Gang Han, Goetz, Giedyk ...). The authors could compare their novel approach to existing methods, thereby highlighting the unique advantages of simultaneous two-photon absorption. Right now the paper looks rather like a PDT or a conventional one-photon catalysis paper when having a look at the cited literature.

Our Response: We appreciate the reviewer's comments and have summarized those recently reported multi-photon-absorbing catalytic systems in the following table, which lists the specific light source and light absorption mechanism of each work. It is apparent that they either require two photon absorption by two different species (consecutive two photon absorption) or follow the triplet-triplet annihilation upconversion mechanism. None of them is like our TPA photocatalysts which are able to directly and simultaneously absorb two photons by one molecule. In addition, those reported photocatalytic systems usually require UV/visible light irradiation or high-power lasers, while our TPA photocatalytic reactions can be driven by inexpensive NIR and low-power LEDs. Nevertheless, these references have been added in the revised manuscript.

Main author	Light source	Light absorption mechanism	Reference
P. S. Francis	447 nm LED	Consecutive two-photon excitation ^a	J. Am. Chem. Soc. 141 , 17646-17658 (2019).
H.-A. Wagenknecht	368 nm LED	Consecutive two-photon excitation ^a	Angew. Chem. Int. Ed. 59 , 300-303 (2020).
M. Goez	532 nm laser	Consecutive two-photon excitation ^a	Angew. Chem. Int. Ed. 53 , 10072-10074 (2014). Chem. Sci. 7 , 3862-3868 (2016). Chem. Eur. J. 24 , 17557-17567 (2018).
M. Giedyk	451 nm LED	Consecutive two-photon excitation ^a	Nat. Catal. 3 , 40-47 (2020).
R. Pérez-Ruiz	445 nm diode laser pointer	Triplet-triplet annihilation upconversion ^b	Chem. Eur. J. 21 , 15496-15501 (2015). Appl. Catal. B: Environ. 237 , 18-23 (2018).
G. Han	720 nm LED	Triplet-triplet annihilation upconversion ^b	J. Am. Chem. Soc. 142 , 18460-18470 (2020). Nat. Commun. 12 , 122 (2021).

^aone photon used to excite the photosensitizer and the second photon is used to excite the in situ generated photocatalyst. ^bTwo sensitizers each absorb one photon to populate their excited states and then transfer their energy to two annihilators, which interact with each other to produce one excited annihilator and another ground-state annihilator.

3) Regarding the DFT calculations presented in the main part of the paper: Rounding the predicted energies up/down (i.e. 551 nm instead of 550.7 nm) might be better to avoid an overestimation of the accuracy of these methods.

Our response: Following the reviewer's comments, we have rounded the predicted energies in the revised manuscript.

4) Page 7: I am not selling Kessil lamps but Kessil LEDs are usually considered as high-power LEDs and not low-power LEDs, given that most commercial LEDs have a much lower output power.

Our response: We have changed "Commercially inexpensive and low-power LEDs" to "Commercially inexpensive LEDs" in the revised manuscript.

5) page 2, "competing absorption by reactants": Perhaps the authors could add side and (catalyst-derived) decomposition products because these species generated in the course of the reaction absorb light as well (to present further advantages of NIR excitation).

Our response: Following the reviewer's comments, we have changed "competing absorption by reactants" to "competing absorption by species involved in the reaction" and cited more references to support this claim, such as *Nat. Catal.* 2020, 3(8), 611-620; *J. Am. Chem. Soc.* 2011, 133(39), 15368-15371; *Proc. Natl. Acad. Sci. U.S.A.*, 2012, 109(39), 15594-15599.

6) There is a typo on the first page of the Supplementary Materials document "Supplemenatry Materials".

Our response: We have corrected this typo.

Reviewer 2

In this work, Y. Sun et al. report a group of ruthenium polypyridyl complexes possessing two-photon absorption (TPA) capability for a variety of photocatalytic organic transformations upon irradiation with NIR light. The authors demonstrate that these two-photon absorption (TPA) photosensitizers can operate for both energy- and electron-transfer reactions. These reactions catalyzed by TPA photosensitizers were systematically studied and the results demonstrate that similar reactivity and selectivity compared to those regular photocatalytic reactions. These observations are not surprising since the same excited states of the photosensitizers were produced by two-photon or one photon excitation. Although authors highlighted TPA photosensitizers present deep penetration into reaction media and negligible competing absorption by organic substrates, this is not the main challenge of photocatalytic organic reactions. Such issue could be easily solved by using flow photoreactor. The utilization of two-photon excitation increases cost and energy consumption. TPA photosensitization is not practically applicable strategy for photocatalytic reactions. It doesn't show clear advantages over the regular photocatalytic reactions. If this type of photosensitizer can be used in biological systems, it might be meaningful. Therefore, the present manuscript doesn't meet the requirement of Nat. Commun.

Our response: We respectfully disagree with the reviewer's conclusion. The reviewer stated that "These observations are not surprising since the same excited states of the photosensitizers were produced by two-photon or one photon excitation." In fact, the "same" excited states of the photosensitizers could be achieved upon two-photon excitation as those under one-photon irradiation is indeed the advantage of our TPA strategy, because it not only utilizes low-energy NIR photons but also produces excited states with competent lifetime and redox potential, in contrast to those one-photon-absorbing NIR photosensitizers which suffer from limited lifetime/redox power even though possess absorption in the NIR region. Currently, most photocatalytic reactions are still driven by UV/visible light, however if one likes to capitalize on solar energy in photocatalysis, the NIR component of the solar spectrum has remained largely underutilized in the most reported photocatalytic reactions. Our TPA strategy provides a direct pathway to utilize the otherwise less-used NIR light in organic synthesis. In addition, NIR photocatalysis exhibits other advantages over UV/visible light in both scale up and fundamental investigation. For instance, a recent *Chemical Review* report (*Chem. Rev.* DOI: 10.1021/acs.chemrev.1c00416) states "*The requirement for detailed understanding of the photophysical requirements, specific light sources and reactor design, provide a significant obstacle to the rapid scale-up of visible-light-mediated reactions. As these challenges are typically the result of poor light penetration into the reaction. A potential alternative to visible light is the use of near-infrared light, which has significantly higher penetration depth.*" Even for a flow reactor, the diameter of the flow tubing will certainly impact the effective absorption of light irradiation. It should be noted that "*only the reaction medium proximal to the vessel wall within 2 mm will experience irradiation in visible light-driven reactions*" (*ACS Cent. Sci.* **2017**, 3, 647-653). Actually, whether flow reactor is the optimal choice for large-scale application is still questionable. Another recent *ACS Central Science* paper has the following statement "*While plug flow reactors maximize light penetration and improve reaction rates on kilograms/day scale, their suitability toward commercial manufacturing is still limited. From an industrial application perspective, the ability to use batch reactors is incredibly advantageous as it does not require specialized equipment and can be easily implemented in any multipurpose facility.*" (*ACS Cent. Sci.* 2020, 6(11), 2053-2059.) With all the considerations in mind, we are confident that that our TPA strategy described in this work represents a breakthrough in organic photocatalysis driven by NIR light irradiation, which is suitable for publication in *Nature Communications*.

Other comments:

1) The molecular structures of Ru complexes 1, 2 and 3 should also be added in figure 1.

Our response: Complex 3 has already been included in Figure 1, while complexes 1 and 2 are control samples which have been included in the synthetic schemes of the supplementary materials, therefore it is not necessary to show them again in Figure 1 of the main text, in order to save space.

2) The authors described that “the utilization of UV/visible photons presents several intrinsic limitations such as poor penetration through reaction media, competing absorption by reactants, and incompatibility against substrates bearing light-sensitive functionalities, as well as limited coverage of the solar spectrum.” Please cite appropriate reference to support your statement.

Our response: We have cited more relevant references to support our claim. These new references are added in the revised manuscript as ref. 8, 9, 10, 11, 12, and 13.

3) Some photocatalytic organic reactions were used to illustrate that TPA photosensitizers can effectively catalyze organic reactions upon irradiation with 740 nm light. The author ought to compare the photocatalytic conversion efficiency of the reaction upon irradiation with single-photon light source.

Our response: Following the reviewer’s comment, we performed a control experiment to compare the conversion efficiencies of a testing reaction, the photocatalytic C-N coupling of benzyl amine, upon irradiation with one-photon (456 nm) or two-photon (740 nm) light source. The Kessil blue LED (PR160L-456, 11.91 W) or NIR LED (PR160L-740-C, 8.18 W) were used as the one-photon or two-photon light source, respectively. **Figure R6a** shows the experimental set-up, in which ten reaction tubes were bundled in one row and the irradiation light was only allowed to excite the reaction tubes from the left side while all the other directions were covered by aluminum foil. The yield of each reaction tube after 6 h irradiation at either 456 or 740 nm was shown in **Figure R6b** and their corresponding ^1H NMR spectra are included in **Figures R7** and **R8**. It is apparent that the one-photon light source (456 nm) was only able to achieve decent yield in the 1st reaction tube while all the remaining tubes exhibited no conversion at all, due to the limited penetration of the 456 nm light. In striking contrast, the two-photon light source (740 nm) was able to not only produce nearly 100% yield in the 1st tube, but also decent yields in the 2nd, 3rd, and 4th tubes. Even the last tube (#10) also produced a yield of 26%. These results unambiguously demonstrate the advantages of NIR light irradiation via two-photon absorption of our photosensitizer.

Figure R6. (a) Experiment set-up. (b) Yields of the photocatalytic C-N coupling of benzyl amine in different reaction tubes. Photocatalysis condition of each tube: 50 mM benzyl amine, 1 mol% **5**, 4 mL CH_3CN , open to air, irradiation at 456 or 740 nm for 6 h at room temperature.

Figure R7. ^1H NMR spectra of the photocatalytic C-N coupling of benzyl amine in different reaction tubes. Photocatalysis condition of each tube: 50 mM benzyl amine, 1 mol% **5**, 4 mL CH_3CN , open to air, irradiation at 456 nm for 6 h at room temperature.

Figure R8. ^1H NMR spectra of the photocatalytic C-N coupling of benzyl amine in different reaction tubes. Photocatalysis condition of each tube: 50 mM benzyl amine, 1 mol% **5**, 4 mL CH_3CN , open to air, irradiation at 740 nm for 6 h at room temperature.

Reviewer 3

This manuscript by Han et al. describes two-photon-absorbing ruthenium polypyridyl complexes and shows that they can promote a variety of photocatalytic organic reactions involving energy transfer to O₂ or redox transformations of the substrate. The work is innovative, well-described, and written in a concise and accessible fashion. The strategy of using two-photon absorption in organic photocatalysis has the potential to take off, provided suitable photosensitizers with large TPA cross sections can be identified, and this paper shows a way to do that with the widely used Ru-bpy catalyst system. I can identify no major technical flaws with the work. I commend the authors for putting together a nice paper, and I recommend publication without extensive revision.

I only have a few minor suggestions for the authors, which are listed below.

1. The authors should cite this recent work that also uses a two-photon strategy for photoredox catalysis: DOI: 10.1021/jacs.1c07617. This paper came out after the authors submitted this present manuscript for consideration, so I don't blame them for not including it and I don't think it detracts from the novelty and impact of their work in a significant way.

Our response: Following the reviewer's comment, we have cited the recommended literature as ref 28 in the revised manuscript.

2. In the experimental section and in the figures showing NMR spectra, ¹³C should be written as ¹³C{¹H} in all cases.

Our response: Following the reviewer's comment, we have changed ¹³C to ¹³C{¹H} in all cases in the revised manuscript.

3. The ¹³C{¹H} and ¹⁹F NMR data is not reported correctly in the experimental section for complex 4. In the ¹³C{¹H} spectrum there should be one or more peaks reported as doublets due to coupling to ¹⁹F; in particular the carbon bound directly to fluorine should have a large coupling constant. In the ¹⁹F spectrum, the peak for PF₆⁻ should be reported as a doublet with coupling to ³¹P, not as two separate peaks.

Our response: We appreciate the reviewer for pointing out those errors. We have corrected the expression for the NMR spectra of complex 4 as follows:

¹³C{¹H} NMR (101 MHz, CD₃CN): 163.0, 158.1, 152.4, 136.0, 133.2 (d, *J* = 3.5 Hz), 130.4 (d, *J* = 8.4 Hz), 125.1 (d, *J* = 34.5 Hz), 121.5, 117.1, 116.8.

¹⁹F NMR (376 MHz, CD₃CN): -72.6 (d, ¹*J*_{P-F} = 704.6 Hz), -112.8.

REVIEWERS' COMMENTS

Reviewer #1 (Remarks to the Author):

The authors carefully revised their manuscript by carrying out several additional experiments and adding numerous references as well as further explanations. The current form of this paper is indeed very convincing and it should be published ASAP.

< In follow-up, private comments to the editorial office, Reviewer 1 states that the experiments performed in response to Reviewer 2's concerns and added citations have adequately prepared this manuscript for publication. >